# Poly-Gamma-Glutamic Acid Functions as an Effective Lubricant with Antimicrobial Activity in Multipurpose Contact Lens Care Solutions

**DOI:** 10.3390/polym11061050

**Published:** 2019-06-16

**Authors:** Chen-Ying Su, Ching-Li Tseng, Shu-Hsuan Wu, Bo-Wu Shih, Yi-Zhou Chen, Hsu-Wei Fang

**Affiliations:** 1Department of Chemical Engineering and Biotechnology, National Taipei University of Technology, 1, Sec. 3, Zhongxiao E. Rd., Taipei 10608, Taiwan; chenying.su@gmail.com (C.-Y.S.); jenny111271@hotmail.com (S.-H.W.); sunda5864@gmail.com (B.-W.S.); 2Graduate Institute of Biomedical Materials and Tissue Engineering, College of Biomedical Engineering, Taipei Medical University, No. 250, Wu-Hsing St., Taipei 110, Taiwan; chingli@tmu.edu.tw (C.-L.T.); max4617@hotmail.com (Y.-Z.C.); 3Institute of Biomedical Engineering and Nanomedicine, National Health Research Institutes. No. 35, Keyan Road, Zhunan Town, Miaoli County 35053, Taiwan

**Keywords:** poly-gamma-glutamic acid, contact lens care solution, antimicrobial activity, lubricating

## Abstract

In order to perform the multiple functions of disinfection, cleansing, and storage, preservatives are often added to contact lens care solutions. The disadvantage of adding preservatives is that this often causes various eye conditions. However, lens care solutions would not be able to disinfect in the absence of such preservatives. In addition, comfort is an important issue for contact lens wearers due to the long periods of time they are worn. It has been shown that lower friction coefficients are correlated with increased comfort. We have previously developed a multipurpose contact lens care solution in which poly-gamma-glutamic acid (γ-PGA) was the main ingredient. In this study, we investigated the antimicrobial activity and lubricating property of our care solution. We showed that there was a synergetic effect of γ-PGA and chlorine dioxide on antimicrobial activity. We also demonstrated that γ-PGA functioned as a lubricating agent. Our results provided evidence of γ-PGA acting as a multi-functional polymer that could be applied in contact lens care solutions.

## 1. Introduction

The number of people wearing soft contact lens’ is increasing, and contact lens care solutions have become the most convenient method of removing deposited components, disinfection, and soft contact lens storage [1]. In order to prolong the shelf life and maintain the disinfecting function of the care solutions after being opened, preservatives are often added. Unfortunately, such preservatives may induce immune reactions that result in eye conditions in the people using the solutions. Contact lens-induced papillary conjunctivitis, contact lens-induced peripheral ulcer, and superior limbic keratoconjunctivitis are common conditions that are caused by the preservatives in contact lens care solutions [2,3,4]. However, preservatives also function as antimicrobial agents that reduce the risk of infection for contact lens wearers. In order to reduce immune reactions and to inhibit microbial growth, developing a contact lens care solution with natural antimicrobial agents might be beneficial to the contact lens wearers. In the eye, *Pseudomonas aeruginosa* (*P. aeruginosa*) has been shown to be one of the most common microorganisms in corneal infections of contact lens wearers [5,6]. In addition, yeasts such as *Candida albicans* (*C. albicans*) have been isolated from contact lenses. Therefore, it is critical for the contact lens care solution to be able to inhibit the growth of *P. aeruginosa* and *C. albicans*.

The contact lens is in contact with the eyelid and the cornea once it is inserted into the eye. The constant blinking causes friction between the eyelid and the lens, as well as between the lens and the cornea. When the contact lens has been worn for an extensive period of time, wearers often feel discomfort that might have resulted from the higher friction coefficient of the contact lens [7,8]. Therefore, providing lubrication becomes an important function of the contact lens care solution.

Poly-gamma-glutamic acid (γ-PGA) is a naturally occurring polymeric biomaterial, mainly derived from *Bacillus anthracis*. It is difficult for proteases to catalyze γ-PGA, which makes it a suitable antibacterial material [9]. It has been demonstrated that the mixture of γ-PGA and chitosan or magnetite nanoparticles can inhibit the growth of *Escherichia coli* (*E. coli*), *Staphylococcus aureus* (*S. aureus*), or *P. aeruginosa* [10]. We have previously shown that γ-PGA can effectively inhibit the growth of *E. coli*, *S. aureus*, and *P. aeruginosa* in a mouthwash [11]. Therefore, we hypothesized that γ-PGA might be a good antimicrobial agent to inhibit the growth of *P. aeruginosa* and *C. albicans*. In addition, we have previously shown that a contact lens solution we developed could increase lubrication for a specific type of contact lens [12]. However, it is unclear whether the lubricating property was due to γ-PGA. In this study, we investigated the antimicrobial and lubricating functions of a multipurpose contact lens solution we developed with γ-PGA as the main ingredient. In addition, we tested the safety of the care solution by conducting in vitro cytotoxicity and in vivo irritation tests. Our results demonstrated that γ-PGA is a multi-functional polymer that can inhibit the growth of certain bacteria and fungi, as well as perform as a lubricant for soft contact lenses.

## 2. Materials and Methods

### 2.1. Chemicals, Reagents, and Contact Lens

The preparation of the multipurpose contact lens care solution is as described below: in 100 mL of distilled water, 0.2 g of ethylenediaminetetraacetic acid (EDTA, Sigma, St. Louis, MO, USA), 0.015 g CaCl_2_ (Sigma, St. Louis, MO, USA), 0.15 g KCl (Sigma), 0.45 g NaCl (Sigma, St. Louis, MO, USA), and 1.8 g Na_2_HPO_4_ (Sigma, St. Louis, MO, USA) were dissolved. Then, 1.5 g of poly-gamma-glutamic acid (γ-PGA, Vedan Enterprise Corporation, Taichung city, Taiwan) and 0.05 g of Poloxamer-407 (Wei Ming Pharmaceutical Mfg. Co., Ltd., Taipei City, Taiwan) were added and filtered through a 0.22 μm filter. Finally, 1 mg of pigallocatechin gallate (EGCG, Sigma, St. Louis, MO, USA), 0.1 mL of hyaluronic acid (HA, Maxigen Biotech Inc., Taoyuan City, Taiwan), and 2.5 μL of 5% chlorine dioxide (ClO_2_, Sigma, St. Louis, MO, USA) were added and mixed well.

For comparing friction coefficient of different solutions, 1 DAY ACUVUE MOIST contact lens (Etafilcon-A, Johnson & Johnson, New Brunswick, NJ, USA) was used. The commercial contact lens care solution used was Hydron Eye Secret Hydration Solution (Yung Sheng Optical Co., Ltd., Taichung City, Taiwan).

### 2.2. Antimicrobial Activity Testing

Standard strains were used in this study: *Pseudomonas aeruginosa* (American type culture collection, ATCC 10145) and *Candida albicans* (ATCC 10231). The microorganisms were added to a centrifuge tube containing 10 mL of broth in the laminar flow hood and cultured at 150 rpm at 37 °C for 18 h. After 18 h, the cultured microorganisms were expanded as required by modifying the culture duration and microbial concentration. Contact lens care solution volumes of 9.9 mL were mixed with 0.1 mL of microbial suspension when the concentration of microorganisms reached 1 × 10^6^ CFU/mL (colony-forming units). Mixtures were made in duplicates. The mixtures were incubated at 35 °C (for *P. aeruginosa* culturing) or 25 °C (for *C. albicans* culturing) for seven days.

On day seven, 1 mL of mixed culture was transferred to 9 mL of Sabouraud Dextrose Broth (SDB) and a series of dilutions were made including 10, 10^2^, and 10^3^ dilutions. Each diluted solution was mixed well and then were sprayed on two plates of Tryptone Soya Agar for *P. aeruginosa* or Sabouraud Dextrose Agar for *C. albicans*. Agar plates were incubated for three days at 35 °C for *P. aeruginosa* cultures or at 25 °C for *C. albicans* cultures. On day three, the numbers of microbial colonies were counted. The antimicrobial activity was calculated using the following formula:

(the initial microbial concentration − the microbial concentration after sevendays)/the initial microbial concentration.
(1)

If the antimicrobial activity was negative, the testing material was considered a non-antimicrobial agent.

### 2.3. Cell Viability Testing

L929 cells (mouse fibroblasts, Food Industry Research and Development Institute, Strains number BCRC 60091) were plated on 96-well culture plates at a density of 10^5^ cells/well in culture medium and cultured at 37 °C overnight. The next day, the culture medium was removed and 100 μL of diluted contact lens care solutions were added (dilution ratio was 0.2 g ± 10%/mL). Triplicates of the negative control group (culture medium), the positive control group (10% Dimethyl sulfoxide, DMSO), and different contact lens care solutions were cultured with L929 cells at 37 °C for 24 h. The next day, the solutions were discarded and 100 μL of MTT (3-(4,5-dimethyl-2-thiazolyl)-2,5-diphenyl-2H-tetrazolium bromide, Sigma 298931, St. Louis, MO, USA) solution was added to each well. In the absence of light, the plate wrapped with aluminum foil was placed in the incubator for 3 h. Then, the MTT solution was discarded and 200 μL of DMSO was added to each well and the plate was shaken gently for 15 min in the dark. When the formazan precipitate was fully dissolved, the solution was transferred to three wells of an Enzyme-Linked Immuno Sorbent Assay (ELISA) plate. The samples were then read by an ELISA plate reader at a wavelength of 570 nm and the optical density (OD) value was obtained. The OD values of each group were averaged and then divided by the averaged OD value of the negative control group. The cell viability was 100% for the negative control group, and a group with cell viability less than 75% was considered as cytotoxic.

### 2.4. Irritation Analysis

Seven male New Zealand White rabbits (weighted around 2.5 to 3.5 kg) with no signs of ocular inflammation or gross abnormalities were used. All experimental procedures were approved by the Institutional Animal Care and Use Committee (IACUC) of Taipei Medical University (IACUC approval number LAC-2015-0317). The animals were housed in standard cages in a light-controlled room at temperature of 23 ± 2 °C, relative humidity of 60% ± 10%, and a 12 h light-dark cycle. Animals were given food and water *ad libitum*. Multipurpose contact lens care solution volumes of 20 μL of were added into each eye, both eyes of each rabbit were administrated with multipurpose solution, and a total of five rabbits were tested (*n* = 10). Two rabbits were administrated with solution without γ-PGA and ClO_2_ as a control (*n* = 4). After adding the multipurpose solution or control solution, the cornea, conjunctiva, and iris of each rabbit were observed and pictures were taken every 24 h for three days. 

### 2.5. In Vitro Contact Lens and Care Solution Friction Testing System

In vitro contact lens and care solution friction tests were performed with a CETR universal micro-tribometer-2 (UMT-2, Bruker, Campbell, CA, USA). The setup of the testing system has been previously described [12]. The contact lens was locked on the upper stage of the UMT-2. The lens was rubbed against a quartz glass which was locked on the lower stage of the UMT-2. A 10 mL volume of PBS, the multipurpose contact lens care solution, the care solution without γ-PGA, or the commercial care solution was added into the lower stage as a lubricant. The contact lens was immersed in the testing solution. The measurement program was set at the normal force of 60 mini-Newtons (mN), the rotation speed at 1 revolution per minute (rpm), and the rotation time was 900 s. The friction coefficient was friction force divided by normal force, and the friction coefficient from the last 600 s was averaged and compared. Each condition was repeated four times.

### 2.6. Statistical Analysis

Differences in friction coefficients or cytotoxicity tests between different experiments were assessed by student’s *t*-test to make allowance of comparisons. A value of *p* < 0.05 was considered significant.

## 3. Results

### 3.1. The Synergetic Effect of γ-PGA and ClO_2_ on Antimicrobial Activity

We tested three bacterial and two fungal strains with various contact lens care solutions and found that *P. aeruginosa* and *C. albicans* were the most difficult microbial strains to inhibit growth. Therefore, we mainly investigated whether γ-PGA and ClO_2_ could be good antimicrobial agents for inhibiting the growth of *P. aeruginosa* and *C. albicans*. We found that when the care solution contained γ-PGA in the absence of ClO_2_, the growth of *P. aeruginosa* and *C. albicans* could not be inhibited efficiently (Table 1), regardless of the concentrations of γ-PGA. In contrast, ClO_2_ could inhibit *P. aeruginosa* very efficiently and could also reduce the growth of *C. albicans* at 1.25 ppm (Table 1).

We further investigated whether there was a better antimicrobial effect if we combined γ-PGA and ClO_2_. When the concentration of γ-PGA was high, it was difficult to dissolve. Therefore, we investigated the antimicrobial activity of 1.5% γ-PGA with various concentrations of ClO_2_. As shown in Table 1, the growth of both *P. aeruginosa* and *C. albicans* was inhibited completely in the presence of 1.25 ppm of ClO_2_ and 1.5% of γ-PGA. Our result indicated that there was a synergetic effect between γ-PGA and ClO_2_ on inhibiting the growth of *P. aeruginosa* and *C. albicans*.

### 3.2. The Multipurpose Contact Lens Care Solution Shows Low Cytotoxicity and Non-Irritation

In order to confirm that the multipurpose contact lens care solution is safe, we conducted cytotoxicity and irritation tests. Since ClO_2_ could inhibit the growth of *P. aeruginosa* at a concentration as low as 0.3125 ppm, we investigated cytotoxicity of solutions with variable ClO_2_. Our results demonstrated that decreasing the concentration of ClO_2_ would reduce cell viability (Figure 1), suggesting our multipurpose contact lens care solution did not cause cytotoxicity.

We then investigated whether the multipurpose contact lens care solution would induce irritation. We found that the solution without γ-PGA and ClO_2_ did not cause irritation, nor did the multipurpose care solution (Figure 2). We observed animals for three days and no irritation occurred, indicating the safety of the multipurpose contact lens care solution (Taipei Medical University IACUC approval number LAC-2015-0317).

### 3.3. Addition of γ-PGA Increases Lubrication of the Multipurpose Contact Lens Care Solution

In order to test whether the multipurpose contact lens care solution could also provide comfort for wearers, we investigated the coefficient of friction using the in vitro friction testing system. The commercial care solution (Hydron) used here was indicated to be a good lubricant, and indeed the friction coefficient of Hydron was lower than that of PBS (Figure 3). The friction coefficient of the multipurpose contact lens care solution was even lower than that of Hydron, suggesting the care solution has a lubricating property. In addition, we tested the friction coefficient of the care solution without γ-PGA. We found that the friction coefficient was increased in the absence of γ-PGA (Figure 3), suggesting that γ-PGA might be functioning as a lubricant.

## 4. Discussion

We investigated the antimicrobial activity and lubricating property of γ-PGA, a polymeric biomaterial, in a multipurpose contact lens care solution in this study. From the two microbial strains we tested, we found that there was a synergetic effect between γ-PGA and ClO_2_ on inhibiting the growth of *C. albicans*. By in vitro friction testing, our results demonstrated that γ-PGA in the multipurpose care solution was a key material for increasing lubrication. In addition, we confirmed our multipurpose contact lens care solution did not cause cytotoxicity or irritation.

Despite the wide applications of γ-PGA in medicine and the food industry, the antimicrobial activity of γ-PGA has not yet been investigated extensively. It has been demonstrated that the derivatives of γ-PGA have antibacterial activity against *Salmonella enteritidis*, *E. coli*, and *S. aureus* [13], γ-PGA had not been shown to be active against fungi. In contrast, ClO_2_ has been demonstrated to be active against multi-drug resistant *P. aeruginosa* [14]. It has been shown that ClO_2_ can inhibit the growth of bacteria by protein denaturation and by involving the covalent oxidative modification of tryptophan and tyrosine [15]. Therefore, it was not surprising when the care solution containing ClO_2_ could display 100% antimicrobial activity for *P. aeruginosa* even at the low concentration of 0.3125 ppm. Some research has shown that ClO_2_ can also damage the plasma membrane of *C. albicans* resulting in K^+^ and ATP leakage rather than disrupting the membrane [16]. This may explain the antimicrobial activity of 1.25 ppm of ClO_2_ against *C. albicans* reaching 97.5% (Table 1). It was not expected that the combination of γ-PGA and ClO_2_ could inhibit the growth of *C. albicans* completely. One possibility of why γ-PGA synergistically enhanced the antifungal activity of ClO_2_ is the hydrophilic and anionic nature of γ-PGA, which may accelerate the leakage of K^+^ and ATP resulting in more severe damage of *C. albicans*. Further investigation of the mechanism of how ClO_2_ and γ-PGA inhibit the growth of *C. albicans* will be required. One potential testing method could be to use another anionic polymer to investigate whether a similar synergetic effect is observed.

In addition, our in vitro friction testing result demonstrated the lubricating property of γ-PGA. γ-PGA has been used in the cosmetic and sanitary industries for its properties that increase the production of natural moisturizing agents such as urocanic acid, pyrrolidone carboxylic acid, and lactic acid [9,17]. Therefore, the higher friction coefficient of soft contact lenses in the solution lacking γ-PGA was expected. It has not been shown previously that γ-PGA can act as a lubricant, and our results provided an additional function of γ-PGA.

γ-PGA is derived from *Bacillus anthracis* and is known to be a biodegradable material. It has also been applied in medicine, especially for use in drug delivery [18,19,20,21], indicating that γ-PGA is biodegradable, edible, and non-toxic. ClO_2_ is an effective biocide [22], but it is uncertain whether it would result in cytotoxicity of human cells. It has been shown that ClO_2_ could kill bacteria within a few minutes of contact, in a short enough time to keep ClO_2_ from penetrating into a wound. or an intact human skin cell [22]. Contact lenses are removed from the eye and rubbed with the multipurpose care solution to clean, and subsequently the lens is rinsed with PBS before putting back into the eye. Therefore, our multipurpose contact lens care solution could sufficiently kill the microorganisms without causing the damage of the eye tissue. Indeed, our cytotoxicity and irritation tests have proved the safety of the multipurpose contact lens care solution. Therefore, our results demonstrated γ-PGA was a non-toxic, biocompatible, and multifunctional material to use in contact lens care solutions.

## 5. Conclusions

The antimicrobial activity and lubricating characteristics of γ-PGA in multipurpose contact lens care solution was investigated in this study. Our antimicrobial test results demonstrated that there was a synergetic effect between γ-PGA and ClO_2_ on inhibiting the growth of *P. aeruginosa* and *C. albicans*. The in vitro friction testing showed that γ-PGA had a lubricating property. Furthermore, the cytotoxicity and irritation tests have confirmed the safety of the care solution. Our studies demonstrated the multi-functional qualities of γ-PGA and provided a potential contact lens care solution formulation that has antimicrobial activity, and a lubricating property, for future applications.

## Figures and Tables

**Figure 1 polymers-11-01050-f001:**
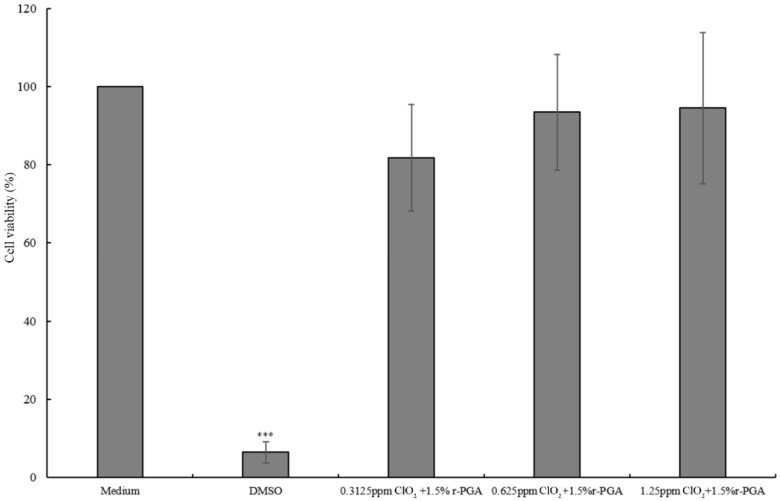
Cytotoxicity testing shows the multipurpose contact lens care solution does not cause cytotoxicity. Cell viability of contact lens care solutions with various concentrations of ClO_2_. *** *p* < 0.001.

**Figure 2 polymers-11-01050-f002:**
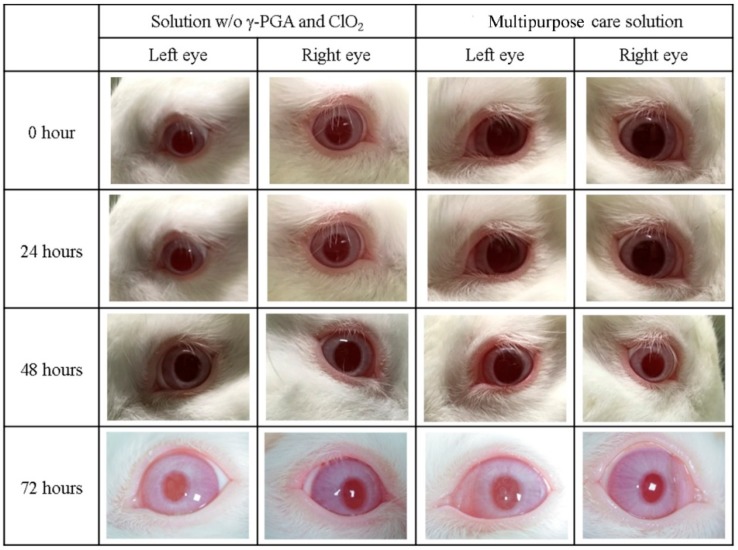
Irritation testing demonstrates the multipurpose contact lens care solution does not cause irritation of rabbit eyes. Contact lens care solution without γ-PGA and ClO_2_, or the multipurpose care solution was administrated on rabbit eyes. Rabbit eyes were observed three times every 24 h. None of the solutions caused any irritation at the time of observation.

**Figure 3 polymers-11-01050-f003:**
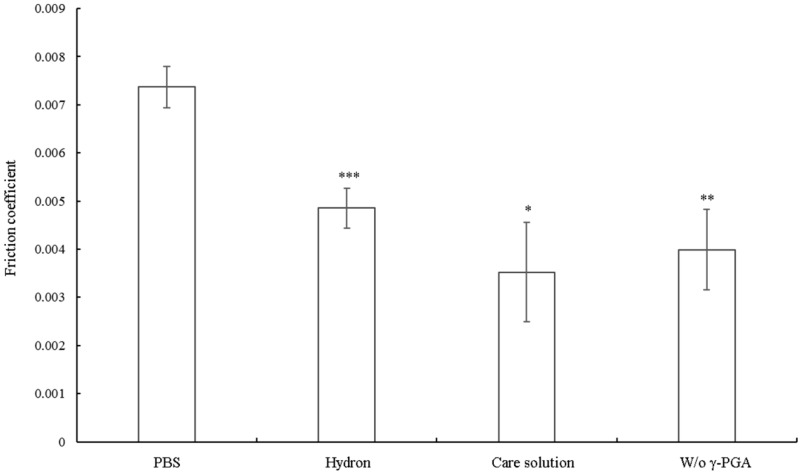
The coefficient of friction of the multipurpose contact lens care solution or the care solution without γ-PGA. When contact lenses were rubbed against the glass in the commercial care solution Hydron, the multipurpose care solution, or the solution without γ-PGA, the coefficients of friction were all lower than that of PBS. The friction coefficient increased in the absence of γ-PGA compared with the solution containing γ-PGA. When comparing with the friction coefficient of PBS, * *p* < 0.05, ** *p* < 0.01, and *** *p* < 0.001.

**Table 1 polymers-11-01050-t001:** The antimicrobial results of contact lens care solutions.

Concentration of ClO_2_ (ppm)	Concentration of γ-PGA (%)	Initial Microbial Concentration (CFU/mL)	Numbers of *Pseudomonas aeruginosa* Colonies after Seven Days	Numbers of *Canidia albicans* Colonies after Seven Days	Antimicrobial activity for *P. aeruginosa*/*C. albicans*
0	7.5	1 × 10^6^ CFU/mL	4.7 × 10^6^	8.4 × 10^5^	N.I./16%
0	2		1.0 × 10^5^	6.0 × 10^6^	90%/N.I.
0	1.5		1.0 × 10^5^	6.0 × 10^6^	90%/N.I.
1.25	0		0	2.5 × 10^4^	100%/97.5%
0.625	0		0	N.A.	100%/N.A.
0.3125	0		0	N.A.	100%/N.A.
1.25	1.5		0	0	100%/100%
0.625	1.5		9.6 × 10^5^	N.A.	4%/N.A.
0.3125	1.5		5.0 × 10^5^	N.A.	50%/N.A.

N.A.: Not applicable, this strain was not tested. N.I.: No inhibition. γ-PGA = poly-gamma-glutamic acid

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
