# Peer review of "Poly-Gamma-Glutamic Acid Functions as an Effective Lubricant with Antimicrobial Activity in Multipurpose Contact Lens Care Solutions"

_polymers, 2019, doi:10.3390/polym11061050_

Round 1
Reviewer 1 Report
The article has a very good quality, good results, but I suggest that is sent to a more specific journal or add characterizations, and more polymer data, to fit better in this journal.
Author Response
Thank you for the encouragement. Poly-gamma-glutamic acid (g-PGA) has been applied as a moisturizer in cosmetics and a bio-control agent in agriculture and gene delivery. However, there is not much research investigating the antimicrobial activity of g-PGA in lubricants. In this manuscript, we focused on characterizing both the antimicrobial activity and lubricating property of g-PGA. Therefore, we consider this manuscript should be a good fit for the Special Issue “Multifunctional Polymeric Biomaterials” of this journal.

Reviewer 2 Report
1. What is the novelty of the present work?
2. Add details of the preparation of the contact lens solution.
3. There is no flow in the introduction, need revision.
4. Why only two strain, P. aeruginosa, and C. albicans was used for the antimicrobial test? What is the reason for choosing this strain?
5. In Fig. 1 why only one results for test solution given. Authors are advised adding the cell viability test of variable concentration (low to high) of lens care solution.
6. Authors are advised to add statistical analysis in Table 1.
7. In line 147, check and correct subheading 3.2.
8. Authors are advised to add comparative results of commercially available contact lens care solution and the prepared solution.
9. There is a lack in the discussion, hence results obtained should be discussed and compared with existing literature.
10. Also, carefully revise the manuscript for typographical and linguistic error.
Author Response
1. What is the novelty of the present work?
Current multipurpose contact lens solutions contain preservatives that trigger immune responses in contact lens wearers, but preservatives also work as an antimicrobial agent. Therefore, it is critical to find suitable antimicrobial agents that do not cause immune responses in order to replace current preservatives. Our work demonstrates that poly-gamma-glutamic acid (g-PGA), a naturally derived polymer, can work as an antimicrobial agent to inhibit the growth of the most common microorganisms.
In addition, our work shows that g-PGA is a lubricant, which is a function that has not yet been demonstrated. Friction between the contact lens, eyelid, and cornea results in discomfort for wearers. The mechanical discomfort will also cause clinical complications. Our results provide evidence that g-PGA could be used as a potential lubricant for contact lens care solutions. Characterizing g-PGA to be an antimicrobial agent as well as a lubricant is the novelty of our work.
2. Add details of the preparation of the contact lens solution.
Thank you for the suggestion. We have edited the preparation of the contact lens solution in section 2.1.
3. There is no flow in the introduction, need revision.
Thank you for the suggestion. We have revised the introduction.
4. Why only two strain, P. aeruginosa, and C. albicans was used for the antimicrobial test? What is the reason for choosing this strain?
We hope to commercialize our contact lens care solution once we prove it to be safe and efficient. Therefore, the antimicrobial activity testing is required to follow ISO (International Organization for Standardization)14730 – contact lens care products, antimicrobial preservative efficacy testing and guidance on determining discard date. According to ISO 14730, Pseufomonas aeruginosa, Staphylococcus aureus, Escherichia coli, Candida albicans, and Aspergillus brasiliensis are used for the antimicrobial test. We have used all five strains to test the antimicrobial activity of our care solution, and it could inhibit the growth of Staphylococcus aureus, Escherichia coli, and Aspergillus brasiliensis in the solution of g-PGA alone, ClO2 alone, or the combination of g-PGA and ClO2. Only the combination of g-PGA and ClO2 could inhibit the growth of P. aeruginosa and C. albicans. Therefore, we only showed the most significant result. In section 2.2, the five strains we used have been mentioned.
5. In Fig. 1 why only one results for test solution given. Authors are advised adding the cell viability test of variable concentration (low to high) of lens care solution.
Thank you for the suggestion. We compared the cell viability of variable concentrations of ClO2 while the concentration of g-PGA was 1.5% (Figure 1). We found that cell viability of 0.3125 ppm ClO2 was lower than at 1.25 ppm. Therefore, the final care solution containing 1.25 ppm ClO2 and 1.5% g-PGA demonstrated the lowest cytotoxicity.
6. Authors are advised to add statistical analysis in Table 1.
Thank you for the suggestion. We added the antimicrobial activity for each material we tested, and edited discussion accordingly.
7. In line 147, check and correct subheading 3.2.
The subheading 3.2 has been edited.
8. Authors are advised to add comparative results of commercially available contact lens care solution and the prepared solution.
Thank you for the suggestion. Because the contact lens care solutions are medical devices, cytotoxicity and irritation testing are required to prove the safety of the products. Therefore, we did not compare the commercial products with our solution in cytotoxicity and irritation testing. Lubrication of contact lens care solutions is not a critical efficacy property, thus we only compared the lubricating property of commercially available contact lens solution and our care solution.
9. There is a lack in the discussion, hence results obtained should be discussed and compared with existing literature.
Thank you for the suggestion. We have edited the discussion.
10. Also, carefully revise the manuscript for typographical and linguistic error.
Thank you for the reminder. The manuscript has been edited by a professional editing agent.

Reviewer 3 Report
In the manuscript, the authors showed a new contact lens care solution composed of poly-gamma-glutamic acid. The care solution with poly-gamma-glutamic acid showed better properties on cytotoxicity, irritation, and lubrication. These results will be helpful and informative In the manuscript, the authors showed a new contact lens care solution composed of poly-gamma-glutamic acid. The care solution with poly-gamma-glutamic acid showed better properties on cytotoxicity, irritation, and lubrication. These results will be helpful and informative for the researchers in the field of polymer chemistry and biomaterials chemistry.
Whereas the reviewer thinks that the authors showed many experimental and analyzed results, some descriptions and discussions are not sufficient. The authors’ manuscript is not suitable for publication in “Polymers” in the present form.
From these considerations, the reviewer recommends accepting for publication in "Polymers," if the following issues are resolved.
(1) The reviewer thinks that poly-gamma-glutamic acid is a new preservative for contact lens care solution. The title of the manuscript should be reconsidered.
(2) How did the authors treat sample contact lens with care solutions for experiments? (wash or sink?) There is no description for this procedure in the experimental section.
(3) Figure 1: What do these concentrations mean, 2% and 77%? An explanation for the control sample should be added.
(4) Figure 2: How was the result of commercial care solution, Hydron?
(5) Is there any precedent study using anionic polymer for contact lens care solutions? The reviewer thinks that some description is required about this.
(6) To consider the synergetic effect of poly-gamma-glutamic acid and ClO2, the reviewer thinks that the experiment using care solution with an anionic polymer instead of poly-gamma-glutamic acid is required for confirming the authors’ consideration.
(7) Is there any precedent study using poly-gamma-glutamic acid as a lubricant? The reviewer thinks that some description is required about this.
Author Response
(1) The reviewer thinks that poly-gamma-glutamic acid is a new preservative for contact lens care solution. The title of the manuscript should be reconsidered.
Thank you for the suggestion. We now called our developed solution as multipurpose contact lens care solution.
(2) How did the authors treat sample contact lens with care solutions for experiments? (wash or sink?) There is no description for this procedure in the experimental section.
Thank you for reminding us. Contact lens was only used in in vitro contact lens and care solution friction testing, and the lens was immersed in the testing solution. We have added a description in section 2.5 for clarification.
(3) Figure 1: What do these concentrations mean, 2% and 77%? An explanation for the control sample should be added.
We have edited figure 1 according to another reviewer’s suggestion, and all the labeling was edited accordingly. The control sample means cells were cultured in medium thus the cell viability was considered as 100%. DMSO was considered as positive control because it causes high cytotoxicity. We then labeled “medium” for 100% cell viability, and labeled “DMSO” instead of positive.
(4) Figure 2: How was the result of commercial care solution, Hydron?
Because the contact lens care solutions are medical devices, cytotoxicity and irritation testing are required to prove the safety of the products. Therefore, we did not conduct irritation testing for the commercial care product, Hydron.
(5) Is there any precedent study using anionic polymer for contact lens care solutions? The reviewer thinks that some description is required about this.
Thank you for the suggestion. The most common polymer used as an antimicrobial agent in contact lens care solution is polyhexamethylen biguanide (PHMB), but PHMB is amphiphilic. There is not any study using an anionic polymer for contact lens care solutions yet.
(6) To consider the synergetic effect of poly-gamma-glutamic acid and ClO2, the reviewer thinks that the experiment using care solution with an anionic polymer instead of poly-gamma-glutamic acid is required for confirming the authors’ consideration.
Thank you for the suggestion. Indeed, using another anionic polymer might be able to confirm our consideration. We have added this possibility on the end of paragraph 2 in the discussion.
(7) Is there any precedent study using poly-gamma-glutamic acid as a lubricant? The reviewer thinks that some description is required about this.
Thank you for the suggestion. There has not been any study using poly-gamma-glutamic acid as a lubricant. We have mentioned this significant finding on the end of paragraph 3 in the discussion.

Round 2
Reviewer 1 Report
Although he did not consider the article in the original submission, believing it would require characterizations and other preliminary results, but because it was considered by the Editor, the manuscript has been significantly improved and now warrants publication in Polymers.